# Circulating Citrate Is Associated with Liver Fibrosis in Nonalcoholic Fatty Liver Disease and Nonalcoholic Steatohepatitis

**DOI:** 10.3390/ijms241713332

**Published:** 2023-08-28

**Authors:** Waseem Amjad, Irina Shalaurova, Erwin Garcia, Eke G. Gruppen, Robin P. F. Dullaart, Alex M. DePaoli, Z. Gordon Jiang, Michelle Lai, Margery A. Connelly

**Affiliations:** 1Division of Gastroenterology, Hepatology, and Nutrition, Department of Medicine, Beth Israel Deaconess Medical Center (BIDMC) and Harvard Medical School, Boston, MA 02215, USA; waseemonline001@gmail.com (W.A.); zgjiang@bidmc.harvard.edu (Z.G.J.); mlai@bidmc.harvard.edu (M.L.); 2Labcorp, Morrisville, NC 27560, USA; shalaui@labcorp.com (I.S.); emgarcia20ub@gmail.com (E.G.); 3Divisions of Nephrology and Endocrinology, University Medical Center Groningen (UMCG), University of Groningen, 9713 Groningen, The Netherlands; e.g.gruppen@umcg.nl (E.G.G.); dull.fam@12move.nl (R.P.F.D.); 4NGM Bio, South San Francisco, CA 94080, USA; adepaoli@ngmbio.com

**Keywords:** citrate, tricarboxylic acid cycle metabolites, liver fibrosis, nuclear magnetic resonance spectroscopy, metabolic dysfunction-associated steatotic liver disease

## Abstract

Nonalcoholic fatty liver disease (NAFLD) is associated with mitochondrial damage. Circulating mitochondrial metabolites may be elevated in NAFLD but their associations with liver damage is not known. This study aimed to assess the association of key mitochondrial metabolites with the degree of liver fibrosis in the context of NAFLD and nonalcoholic steatohepatitis (NASH). Cross-sectional analyses were performed on two cohorts of biopsy-proven NAFLD and/or NASH subjects. The association of circulating mitochondrial metabolite concentrations with liver fibrosis was assessed using linear regression analysis. In the single-center cohort of NAFLD subjects (n = 187), the mean age was 54.9 ±13.0 years, 40.1% were female and 86.1% were White. Type 2 diabetes (51.3%), hypertension (43.9%) and obesity (72.2%) were prevalent. Those with high citrate had a higher proportion of moderate/significant liver fibrosis (stage F ≥ 2) (68.4 vs. 39.6%, *p* = 0.001) and advanced fibrosis (stage F ≥ 3) (31.6 vs. 13.6%, *p* = 0.01). Citrate was associated with liver fibrosis independent of age, sex, NAFLD activity score and metabolic syndrome (per 1 SD increase: β = 0.19, 95% CI: 0.03–0.35, *p* = 0.02). This association was also observed in a cohort of NASH subjects (n = 176) (β = 0.21, 95% CI: 0.07–0.36, *p* = 0.005). The association of citrate with liver fibrosis was observed in males (*p* = 0.005) but not females (*p* = 0.41). In conclusion, circulating citrate is elevated and associated with liver fibrosis, particularly in male subjects with NAFLD and NASH. Mitochondrial function may be a target to consider for reducing the progression of liver fibrosis and NASH.

## 1. Introduction

Nonalcoholic fatty liver disease (NAFLD) (newly proposed nomenclature: metabolic dysfunction-associated steatotic liver disease or MASLD) is one of the most common causes of chronic liver disease [1,2]. An international consensus published in 2020 defined metabolic dysfunction-associated fatty liver disease (MAFLD) as hepatic steatosis as evidenced by histological (liver biopsy), imaging, or blood-based biomarker evidence in addition to one of the following three criteria: overweight/obesity, type 2 diabetes (T2D), or evidence of metabolic dysregulation [3]. The new definitions for MAFLD and MASLD removed references to alcohol usage, largely because the term “nonalcoholic” did not accurately capture the true etiology of the disease [2,3] The term MASLD takes it one step further by removing the word “fatty” which has been thought by some clinicians and patients to be somewhat stigmatizing [2]. The natural history of NAFLD/MASLD ranges from benign steatosis to nonalcoholic steatohepatitis (NASH), liver fibrosis, cirrhosis or hepatocellular carcinoma [4]. Due to the high global prevalence of NAFLD, it is one of the primary pathologies leading to liver insufficiency requiring liver transplantation [5]. Still, the mechanisms of NAFLD progression from steatosis to NASH with advanced fibrosis are not well established. It was previously recognized that NAFLD is associated with impaired fatty acid oxidation and mitochondrial damage contributing to oxidative stress [6]. Increased free fatty acid uptake and β-oxidation, as well as partial uncoupling of the mitochondrial electron transport chain, causes leakage of electrons which leads to free radical-induced hepatocyte injury followed by inflammation and fibrosis (Figure 1) [7,8,9]. Mitochondrial damage and intra-hepatocyte fatty acid accumulation in NAFLD can lead to dysfunction of the citric acid or tricarboxylic acid (TCA) cycle [9,10]. This premise is further supported by observations that metabolites of mitochondria activity, such as circulating citrate and intrahepatocyte β-hydroxybutyrate, may be elevated in NASH [9,11].

Metabolic syndrome (MetS) and NAFLD are associated with higher circulating free fatty acid levels and elevated glucose [12], both of which are catabolized to acetyl CoA before entrance into the TCA cycle. Isocitrate dehydrogenase is a rate-limiting step in this pathway and high substrate concentrations lead to increased levels of isocitrate which in turn increases citrate levels, largely because the aconitase-catalyzed conversion from citrate to isocitrate is reversible [11]. With the exception of some ketogenic diets, high-fat diets and MetS both confer aconitase inhibition which conceivably contributes to the elevation of circulating citrate [13,14,15]. Citrate forms a complex with iron and this complex, in combination with hydrogen peroxide, causes oxidative stress [16]. Another rare cause of elevated citrate levels is citrin deficiency, which is characterized by aspartate/glutamate carrier deficiency and upregulation of mitochondrial citrate/isocitrate carrier (CiC, also known as SLC25A1) leading to citrin deficiency [11,17]. Furthermore, it has been observed that citrin deficiency is associated with NAFLD [18].

Advancing fibrosis, one of the major characteristics of NAFLD disease severity and progression, has been linked to mitochondrial impairment [19,20]. Notably, the relation of circulating mitochondria-related metabolites (e.g., citrate, pyruvate and ketone bodies) with liver fibrosis, as determined by liver histology, has not been established. We hypothesized that circulating mitochondrial metabolites may be biomarkers for the progression of liver fibrosis in the context of NAFLD and/or NASH. The aim of this study, therefore, was to examine the relationship between circulating citrate, pyruvate and ketone body levels with liver fibrosis in NAFLD.

## 2. Results

### 2.1. Baseline Characteristics

In the Beth Israel Deaconess Medical Center (BIDMC) NAFLD registry, 187 patients had a liver biopsy within 3 months of the blood collection that was used for the measurement of serum citrate levels. The prevalence of comorbidities was higher among patients with NAFLD as compared to controls (Table 1). In addition, fasting glucose and TG were higher and HDL-C was lower in the subjects with NAFLD compared to controls. Liver fibrosis stages F2–F4 were observed in 45.5% of the subjects in the NAFLD cohort and 7.5% had cirrhosis (Table 1).

Of the three mitochondrial metabolites, citrate and pyruvate concentrations did not differ between the control group and subjects with metabolic disease (*p* = 0.54 and 0.33, respectively), but tended to be higher in subjects with NAFLD than in controls (*p* = 0.05 and 0.001, respectively). Citrate and pyruvate concentrations were significantly higher in subjects with NAFLD compared to those with metabolic disease (*p* = 0.009 and <0.0001, respectively) (Table 1). On the other hand, total ketone body levels were higher in subjects with metabolic disease (*p* = 0.008), but even higher in subjects with NAFLD compared to controls (*p* = 0.001). However, there was no significant difference in the concentration of ketone bodies between subjects with NAFLD and those with metabolic disease (*p* = 0.65). Despite being higher than normal, the mean ketone body concentration in the NAFLD cohort (0.23 mmol/L) was much lower than one would experience with diabetic ketoacidosis (>3 mmol/L). Figure 2 shows the individual levels of citrate, pyruvate and total ketone bodies according to the degree of fibrosis. In this univariable analysis, citrate (rho = 0.23, *p* = 0.002) and pyruvate (rho = 0.15, *p* = 0.04) but not total ketone bodies (rho = 0.12, *p* = 0.12) were associated with the degree of liver fibrosis. The association of citrate with the degree of fibrosis was significant in males (rho = 0.31, *p* = 0.008) but not in females (rho = 0.15, *p* = 0.23).

### 2.2. Citrate Levels and Fibrosis in the NAFLD Cohort

Of the 187 patients in the NAFLD cohort, 38 patients (20.3%) had citrate levels >150 μM. The comparison of patients with normal range citrate (≤150 μM) [21] and those with high citrate levels >150 μM is provided in Table 2. Subjects with high citrate levels were older and more likely to be female. Aspartate transaminase (AST) levels were higher, whereas total cholesterol and low-density lipoprotein cholesterol (LDL-C) levels were lower in the high citrate group. Patients with high citrate (>150 μM) also had higher NAFLD disease activity (NAS). A higher percentage had moderate liver fibrosis (F ≥ 2) as well as advanced liver fibrosis (F ≥ 3), while cirrhosis (F4) prevalence was not statistically different between subjects with low vs high citrate levels, although the difference in the prevalence in cirrhosis may not have reached significance due to the low numbers. Circulating pyruvate and ketone body levels were higher in the high citrate group (Table 2).

Citrate levels were higher in stage F2–F4 fibrosis as compared to F0-F1 fibrosis (128.0 ± 43.6 vs. 116.3 ± 35.3 µM, *p* = 0.045) in the total NAFLD cohort (n = 187). Citrate was numerically higher in females as compared to males (127.9 ± 45.9 vs. 117.4 ± 34.4 µM, *p* = 0.07). Univariable logistic regression analysis showed that an increase in 1 SD of citrate (β = 0.27, 95% CI: 0.10–0.45, *p* = 0.003) and pyruvate levels (β = 0.18, 95% CI: 0.001–0.36, *p* = 0.049) were associated with higher fibrosis stage, whereas an increase in 1 SD of ketone bodies was not associated with higher liver fibrosis stage (β = 0.08, 95% CI: −0.10–0.26, *p* = 0.36) (Table 3).

The association of citrate with higher liver fibrosis remained statistically significant when adjusted for age, sex, (Model 1: β = 0.27, 95% CI: 0.08–0.45 per 1 SD increase, *p* = 0.004), MetS (Model 2: β = 0.20, 95% CI: 0.02–0.38 per 1 SD increase, *p* = 0.03) and NAS (Model3-β = 0.19, 95% CI: 0.03–0.35 per 1 SD increase, *p* = 0.02) (Table 4).

Sensitivity analysis (using ordinal logistic regression analysis) showed that higher citrate was associated with higher fibrosis stage independent of age, sex, MetS and NAS (OR: 1.41, 95% CI: 1.03–1.93 per 1 SD increase, *p* = 0.03). This association was statistically significant for differentiating early-stage from moderate liver fibrosis (stage F0/F1 vs. F2–F4) but not significant for differentiating earlier stages from advanced fibrosis (F3/F4) or cirrhosis (F4) (Appendix A).

A secondary analysis was performed to examine the association of citrate with liver fibrosis separately in males and female subjects. The sex-specific analysis showed that an association of citrate with an increasing degree of liver fibrosis was observed in males (β = 0.34; 95% CI: 0.10–0.57, per 1 SD increase *p* = 0.005), whereas this association was not seen in females (β = 0.09, 95% CI: −0.13–0.32 per 1 SD increase, *p* = 0.41) (Table 5).

### 2.3. Confirmation of the Relationship between Citrate Level with Liver Fibrosis

In an independent cohort of subjects with biopsy-proven NASH (n = 176), mean citrate level was higher in stage F2–F4 fibrosis as compared to stage F0-F1 fibrosis (119.2 ± 30.8 vs. 109.8 ± 29.9 µM, *p* = 0.04). An increase in citrate was positively associated with liver fibrosis in univariable analysis (β = 0.23, 95% CI: 0.08–0.37 per 1 SD increase, *p* = 0.002). This association remained significant after adjusting for age, sex, available MetS components (high TG, high glucose and low HDL-C), and NAS (β= 0.21, 95% CI: 0.07–0.36, *p* = 0.005) (Table 6). Consistent with the results obtained in the NAFLD cohort, in multivariable analysis males had a statistically significant association of citrate with higher liver fibrosis (β = 0.23, 95% CI: 0.07–0.40 per 1 SD increase, *p* = 0.005); which was not observed in females (β = 0.09, 95% CI: −0.13–0.32 per 1 SD increase, *p* = 0.29).

### 2.4. Association of Pyruvate and Total Ketone Bodies with Liver Fibrosis in NAFLD

Multivariable analysis adjusted for age and sex showed a modest association between pyruvate levels with higher liver fibrosis (β = 0.16, 95% CI −0.002–0.35, per 1 SD increase, *p* = 0.05) but this association was attenuated after further adjustment for MetS and NAS (Appendix A). The ketone bodies’ association with liver fibrosis in NAFLD was insignificant in all multivariable models (Appendix A).

## 3. Discussion

The current study strengthens the previously reported observation that circulating citrate levels were elevated in a small group of six NAFLD patients as compared to healthy subjects [11]. Our observations in multiple larger cohorts extend those of the previous report by further identifying histology-graded liver fibrosis as a potential pathophysiological connection between higher citrate, mitochondria impairment, and NAFLD/NASH progression. We found that higher citrate levels were associated with moderate and advanced fibrosis but were not further elevated in cirrhosis. This would raise the possibility that with disease progression, hepatocytes undergo apoptosis blunting a further increase in citrate levels. Furthermore, the association between serum citrate and liver fibrosis was only observed in males but not in females. This is not totally unexpected as sex-specific metabolic differences in subjects with NAFLD have been noted previously in both mice and humans [22]. Our observations were made in a well-characterized NAFLD registry and confirmed in an independent cohort of subjects with biopsy-proven NASH, indicating the robustness of these findings.

We did not observe elevations in plasma citrate in the metabolic disease cohort in which we tried to reduce the chance of including participants with latter-stage NAFLD by excluding participants with liver enzyme elevations. This observation suggests that the increased circulating citrate in subjects with NAFLD may not be due to metabolic disease per se but is more likely related to the stage of liver disease. Previous studies in animal models suggest that increased stellate cell TCA cycle activity in steatohepatitis leads to elevated circulating citrate levels [22]. Initially, increased fatty acid beta-oxidation leads to an increase in circulating ketone bodies, as observed in our study. However, further increases in acetyl CoA influx into the TCA cycle may cause inefficient β-oxidation and free fatty acid disposal [22]. This will eventually blunt ketogenesis which can culminate in the production of toxic free fatty acid metabolites (e.g., diacylglycerol, ceramides) and lipotoxicity, followed by inflammation and further mitochondrial impairment [22]. In the liver, this scenario may contribute to the transition from benign steatosis to advanced fibrosis [23,24]. This was clearly shown in a publication by Sunny et. al., who showed that mitochondrial oxidative metabolism is greater in subjects with NAFLD despite higher circulating ketone body concentrations [25]. Patients with NAFLD in this study had higher rates of lipolysis, gluconeogenesis, and intrahepatic TG content, which led to oxidative stress and liver damage [25].

In agreement with previously published data, in the current study females had slightly higher circulating citrate levels [26]. The association between citrate levels and liver fibrosis, however, was observed in males but not females. These sex-specific differences are not completely understood, but sex differences in NAFLD have been noted previously in both mice and humans [22]. Males are more likely to have central obesity whereas pre-menopausal females are more likely to have subcutaneous obesity and the former have a higher risk of chronic liver inflammation and advanced liver fibrosis [22]. This leads to a higher incidence of hepatic tumors in males than in females [22]. Further, it has been shown by computer modeling that male and female livers are metabolically distinct with unique regulatory molecules modulating sex-specific metabolism and outcomes [22]. Along these lines, gender dimorphism has been observed for liver pyruvate kinase (L-PK expression [27]. L-PK was over-expressed in males which was associated with a higher influx of pyruvate into mitochondria leading to increased TG storage through enhanced de novo lipogenesis [27]. Additionally, animal studies have shown higher mitochondrial DNA levels in male NAFLD/NASH mice, suggesting sex-specific differences in mitochondrial metabolism and pathologies in NAFLD [22]. A large study of over 50 cardiometabolic traits in 100 inbred strains of mice revealed sex differences in mitochondrial function that were related to genetic differences [28]. Results showed reduced mitochondrial function in males compared to females, which was associated with increased susceptibility to obesity as well as insulin resistance [28]. Moreover, citrate synthase in myocardial mitochondria declines with age and is subject to dietary effects in male but not female rats [29]. Gender dimorphism in glycolytic and mitochondrial metabolic pathways has also been observed in myocardial mitochondria from aging primates [30]. Based on the results of the current study, it is possible that liver disease progression has differential responses to disruption of beta-oxidation and impaired TCA cycle in males as compared to females.

Citrate has been shown to be associated with incident mortality [26,31]. In a study with 17,345 individuals, citrate was one of four NMR-measured metabolites that were shown to predict cardiovascular mortality, death from cancer, and all-cause mortality [31]. Citrate is also part of a multimarker algorithm, called the Metabolic Vulnerability Index (MVX) which portends mortality risk [26]. Citrate is the part of MVX that is related to metabolic dysfunction and malnutrition and MVX scores were shown to be associated with risk of mortality in two independent cohorts of high-risk cardiovascular disease patients [26]. Higher MVX scores have also been shown to be associated with a higher risk of liver-related mortality, hepatic decompensation, and estimated glomerular filtration rate (eGFR) decline in subjects with NAFLD even when accounting for age, race, sex, ethnicity and fibrosis stage [32]. Given that fibrosis stage is directly related to liver-related outcomes and mortality in patients with NAFLD [33,34], the observations that citrate is related to liver fibrosis in NAFLD in our study and that MVX, which includes citrate, is related to mortality in patients with NAFLD, are intriguing.

Citrate serves as the precursor of lipogenesis, as it provides acetyl CoA for fatty acid synthesis. Citrate also reduces glycolysis by inhibiting phosphofructokinase-1 and -2 and increases gluconeogenesis by activating fructose 1,6-bisphosphatase and acetyl CoA carboxylase alpha [35,36]. Inhibition of citrate carrier by CTPI-2 in mice enables them to tolerate a high-fat diet over a prolonged period without developing obesity, disruption in glucose homeostasis and significant liver damage. CTPI-2 may thus serve as a glucose-controlling agent which suggests the potential benefit of using the TCA cycle as a target for pharmacological treatment to prevent the progression of steatosis to NASH [37]. It is hard, however, to tell for sure if the increase in circulating citrate, which is presumably caused by the increase in substrates for the TCA cycle, is simply a consequence of obesity, T2D and/or NAFLD or if it contributes to the loss in mitochondrial homeostasis. Additionally, because citrate can penetrate the sarcolemma and is a fundamental substrate for cellular metabolism in several different energy processes, there is evidence that taking sodium citrate as a dietary supplement could be beneficial [38]. However, it is unknown whether higher citrate levels represent a beneficial adaptation or if citrate may exacerbate the underlying disease state. More work needs to be conducted to understand the cause and effect of these observations.

Strengths of our study include a comprehensive evaluation of the relationship between circulating citrate levels and liver fibrosis in a well-characterized biopsy-proven NAFLD cohort. Moreover, our observations were confirmed in a fairly large cohort with biopsy-proven NASH. Several limitations of the study should be noted. The cross-sectional study design precludes the ability to determine causality. The BIDMC NAFLD cohort, the Metabolic Disease cohort and the NASH clinical trial were comprised of predominantly White subjects; therefore, these results do not necessarily apply to other races/ethnicities. Future studies to confirm these results should be conducted in more diverse populations. The proportion of subjects with stage F3 and F4 liver fibrosis in the NAFLD cohort was rather low which may have caused the analyses for these categories to be underpowered. Some of the limitations in the BIDMC registry, however, were mitigated by the validation in a second NASH cohort with more advanced liver fibrosis where we were able to reproduce similar results. Our study does not explain the mechanism of circulating citrate and liver fibrosis association but does imply that advancing liver fibrosis may be tied to progressive mitochondrial impairment. Decompensated cirrhosis cases were not included in our study and results may be different in these patients. There can be unmeasured confounders that can affect this association including dietary habits, physical activity, and medication history.

## 4. Materials and Methods

### 4.1. Single-Center NAFLD Cohort

A total of 187 patients from a prospectively followed NAFLD Registry enrolled at BIDMC, Boston, MA, USA, were selected for this study [39]. Patients underwent a liver biopsy at study enrollment. Those with viral liver diseases, significant alcohol use (>21 units per week for males and >14 units per week for females, with 1 unit assumed to contain 10 g of alcohol) [40], genetic disorders, drug-induced liver injury and biliary disorders were excluded. Liver disease etiologies were identified based on laboratory studies, and clinical, radiological and/or histological information. History of alcohol use was recorded by interviewing patients and family members. The clinical protocol for the registry was approved by the BIDMC Institutional Review Board (IRB) (#2009P000301). The study was conducted in accordance with the Declaration of Helsinki guidelines and all patients gave written informed consent at the time of enrollment. Most of the laboratory and clinical data were recorded within 7 days of the liver biopsy. Anthropometric measurements including height, waist circumference, blood pressure and body mass index (BMI), which was calculated as weight divided by height squared, were obtained at the index visit. Standardized laboratory tests included basic chemistry, liver function panel, standard lipid profile, fasting glucose, and glycated hemoglobin (HbA1c). Serum bilirubin and liver enzymes were quantified using spectrophotometry [40,41]. Nuclear magnetic resonance (NMR) spectra were acquired in 2014 and 2016 from fasting serum samples as previously described [39].

MetS was defined, based on the National Cholesterol Education Program Adult Treatment Panel III (NCEP ATP III) guidelines, as three or more of the following risk determinants: (1) fasting blood glucose ≥ 110 mg/dL (glycated hemoglobin/HbA1c ≥ 5.7% was also considered as elevated glucose equivalent), (2) hypertension (HTN) and/or use of anti-hypertensive medication, (3) low high-density lipoprotein cholesterol (HDL-C < 40 mg/dL for males and <50 mg/dL for females), (4) high triglycerides (TG ≥ 150 mg/dl) and (5) waist circumference ≥ 102 cm for males and ≥88 cm for females [42,43]. T2D was defined as fasting glucose ≥ 126 mg/dL, current use of anti-diabetic drug or HbA1c ≥ 6.5% [44].

Liver biopsies were evaluated by at least two hepatobiliary pathologists at BIDMC. NASH cases were diagnosed based on guidance published by the American Association for the Study of Liver Disease (AASLD) NASH Task Force [45]. Disease severity was measured as NAFLD activity score (NAS, 0 to 8), calculated from hepatic steatosis, lobular inflammation, and balloon degeneration [45,46]. The stage of fibrosis was graded from F0 to F4 as follows: stage F0 was the absence of fibrosis, stage F1 was perisinusoidal or portal fibrosis, stage F2 was perisinusoidal and portal or periportal fibrosis, stage F3 was septal or bridging fibrosis and stage F4 was cirrhosis [47]. Definitions for liver fibrosis stages in this study include significant fibrosis (F1, F2, F3), advanced fibrosis (F3, F4) and cirrhosis (F4).

### 4.2. Metabolic Disease Cohort (n = 132)

Patients with T2D and/or MetS who were >18 years old participated after written informed consent had been obtained [48]. T2D was diagnosed by the participant’s primary care physician based on established criteria. All of the participants in this cohort were white. Patients using insulin were excluded from the study, but the use of anti-hypertensive medication was allowed. Patients who smoked and subjects who were using lipid-lowering drugs were also excluded, as were participants with the following: liver enzymes > 2 times the upper reference interval (to reduce the chance of including participants with later stage NAFLD), a history of cardiovascular disease, chronic kidney disease (defined as eGFR < 60 mL/min/1.73 m^2^ and/or albuminuria), or known thyroid dysfunction. NMR spectra were acquired in 2013 from fasting serum samples collected between 2003 and 2004. After the exclusion of subjects with missing NMR or covariate results, 132 subjects were analyzed. The clinical study protocol was approved by the medical ethics committee (METC) of the University Medical Center Groningen (METC02.164; approval date 4 October 2002), The Netherlands, and the study was conducted in accordance with the Declaration of Helsinki guidelines. Clinical procedures and laboratory measurements were described in detail previously [48].

### 4.3. Control/Referent Cohort (n = 98)

Healthy control participants were selected from a study performed at LipoScience, Raleigh, NC, USA (now Labcorp). This population included adults between 18 and 84 years of age. Those with a history of coronary artery disease, chronic kidney disease, heart failure, T2D, hypertension and obesity (BMI ≥ 30 kg/m^2^) were excluded. In addition, samples for the healthy cohort were chosen to be age and gender-matched with the Metabolic Disease Cohort and the BIDMC NAFLD Registry for a total of 98 subjects. NMR testing of serum samples was performed in 2015 and the stored NMR spectra were used to quantify mitochondrial metabolites including citrate, pyruvate and ketone bodies. Additional details for this apparently healthy population have been previously reported [49]. The study protocol was approved by the Chesapeake IRB, North Carolina, USA (Pro00001317, 2012). This study was conducted in accordance with the Declaration of Helsinki guidelines and all of the patients provided written informed consent.

### 4.4. NASH Cohort (n = 176)

Baseline samples and data from an independent biopsy-proven, multicenter, NASH clinical trial were used to replicate the current findings. Subjects were enrolled by 18 gastroenterology centers in Australia and the USA. Adults aged 18 to 75 years, with biopsy-proven NASH, based on the NASH clinical research network (CRN) histological scoring system, were enrolled [47]. Those with a minimum NAFLD activity score (NAS) of 4, stage F1, F2 or F3 liver fibrosis and liver fat content ≥8% (assessed by magnetic resonance imaging-proton density fat fraction) were included. Those with acute or chronic liver disease unrelated to NAFLD (e.g., drug-induced, excessive alcohol use), a history of decompensated or compensated cirrhosis, liver transplantation, a cardiovascular event within 6 months of screening and those with type 1 diabetes were excluded. NMR spectra were acquired in 2016 from fasting EDTA plasma samples collected between 2015 and 2016. The study protocol was approved by local ethics committees (IRB numbers and clinical trial sites are available upon request). The study was conducted in accordance with the Declaration of Helsinki, International Conference on Harmonization and E6 Good Clinical Practice and all patients provided informed consent. This cohort has been previously described in the literature and is listed in ClinicalTrials.gov (NCT02443116) [50].

### 4.5. NMR-Mediated Quantification of Citrate, Pyruvate and Total Ketone Bodies

Citrate levels were measured using NMR spectroscopy at Labcorp (Morrisville, NC, USA). The method for quantifying citrate from digitally stored NMR spectra, previously collected from serum or EDTA plasma samples, was described in detail [49]. The stability of citrate has been established in samples that were frozen for up to 12 years at <−70 °C. Coefficients of variation (%CV) for inter-assay precision for NMR-measured citrate ranged from 5.2% for a high to 9.6% for a low-concentration pool. Citrate ≥50 µM was considered for analysis and citrate >150 µM was defined as high citrate, samples with citrate <50 µM were excluded [21]. The detailed NMR method for quantifying total ketone bodies (β-hydroxybutyrate, acetoacetate, acetone) and assay performance for the ketone body assay was published previously [51]. Pyruvate was quantified similarly to both citrate and the ketone bodies by deploying a non-negative least square deconvolution algorithm. When the NMR spectra are collected the same as they are for the NMR LipoProfile^®^ Test [49,51], pyruvate is a single NMR signal peak at 2.33 pm. The pyruvate peak integral was measured and results were calculated. The conversion factor for converting pyruvate concentration from units to µM was determined using a spiking experiment whereby known pyruvate concentrations were added to the dialyzed serum.

### 4.6. Statistical Analysis

Baseline characteristics are presented as mean (±SD) for continuous variables and as frequencies for categorical variables and were compared between normal and elevated citrate levels (>150 µM) [21]. Baseline characteristics were further compared between the NAFLD cohort and an external cohort of apparently healthy adults. Continuous variables were compared using Student’s *t*-test and categorical variables by χ-square test. Simple relationships of plasma citrate, pyruvate and total ketone bodies and the degree of fibrosis were assessed using Spearman correlation coefficients (rho). For univariable and multivariable modeling, Z-scores for citrate, pyruvate and ketone body levels were calculated by the ratio of difference of each variable and its mean over standard deviation. Z-scores were used in the main models to determine coefficients per 1 standard deviation (SD) increase. Adjusted associations of citrate levels (expressed per 1 SD increment) with liver fibrosis stage were calculated using multivariable linear regression analysis. Covariates in multivariable linear regression analysis were identified based on the biological relevance of being a potential confounder of the relationship. Model 1 was adjusted for age and sex. Model 2 was adjusted for age, sex and MetS components (elevated glucose, HTN, high waist circumference, low HDL-C and high TG) and model 3 was adjusted for age, sex, components of MetS and liver disease progression index or the NAFLD activity score (NAS). Sensitivity analysis was performed using ordinal logistic regression. Intermediate category logistic regression was conducted to test the assumption of proportional odds. Subgroup analyses were performed in non-cirrhotic, males and females separately. The association of pyruvate and ketone bodies with liver fibrosis was assessed using univariable and multivariable models. The multivariable models for the association of citrate with the severity of liver fibrosis were also tested in a NASH validation cohort. Statistical analyses were conducted using STATA 17.0 and SAS 9.4. All tests were two-tailed and a *p*-value of <0.05 was considered significant.

## 5. Conclusions

In conclusion, we documented for the first time an association of increased circulating citrate with liver fibrosis in NAFLD and NASH patients. This observation may provide pathophysiological insight into the potential involvement of the TCA cycle in NAFLD disease progression. Additional metabolites, such as α-ketoglutarate and succinate should be studied in order to fully understand how mitochondrial impairment contributes to NAFLD/NASH disease progression. Future studies should focus on understanding the mechanism that underscores these observations and investigating if mitochondrial function may be a viable target for future therapies to reduce liver fibrosis progression in patients with NAFLD/NASH.

## Figures and Tables

**Figure 1 ijms-24-13332-f001:**
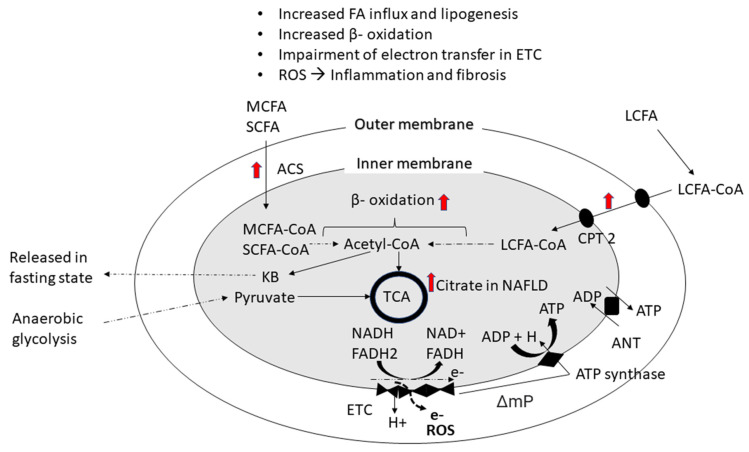
Disrupted fatty acid metabolism in NAFLD. The figure shows fatty acid oxidation and energy production in the mitochondria. SCFA and MCFA freely enter mitochondria whereas LCFA enters through carnitine palmitoyltransferase. These fatty acids are converted to acetyl CoA via beta-oxidation. Ketone bodies are generated during fasting, and pyruvate enters the mitochondria after anaerobic glycolysis. Fatty acid oxidation generates NADH and FADH2 which transfer electrons (e−) to the electron transport chain (ETC). Flow of e− through ETC is coupled with release of protons in intermembrane space which generates membrane potential. This membrane potential is utilized in phosphorylation of ADP to ATP which is then released from matrix. Impairment of ETC causes release of electrons and formation of reactive oxygen species (ROS). Abbreviations: ACS, acetyl coenzyme A synthase; ADP, adenosine diphosphate; ANT, adenosine nucleotide translocator; ATP, adenosine triphosphate; CPT, carnitine palmitoyltransferase; ETC, electron transport chain; FAD, flavin adenine dinucleotide; KB, ketone bodies; LCFA, long chain fatty acid; MCFA, medium chain fatty acid; ΔmP, mitochondrial membrane potential; NAD, nicotinamide adenine dinucleotide; ROS, reactive oxygen species; SCFA, short chain fatty acid; TCA, tricarboxylic acid cycle or citric acid cycle.

**Figure 2 ijms-24-13332-f002:**
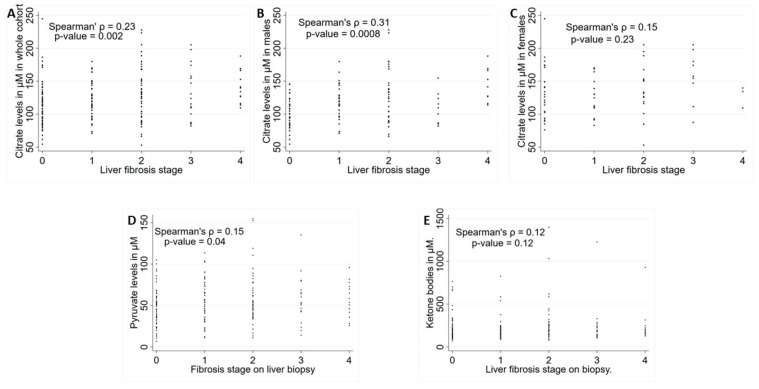
(**A**) citrate in the NAFLD cohort (n = 187), (**B**) citrate in males (n = 112), (**C**) citrate in females (n = 75), (**D**) pyruvate in the NAFLD cohort (n = 187) and (**E**) total ketone bodies in the NAFLD cohort (n = 187) according to the degree of fibrosis (F0: absent through F4: cirrhosis). Spearman correlation coefficients (rho) and *p*-values are noted.

**Table 1 ijms-24-13332-t001:** Baseline characteristics of a Control Cohort (age, gender-matched), Metabolic Disease Cohort and BIDMC NAFLD Cohort.

Variables	Control Cohort (n = 98)	Metabolic Disease Cohort (n = 132)	BIDMC NAFLD Cohort (n = 187)	Control vs. Metabolic Disease *p*-Value	Control vs. NAFLD *p*-Value	Metabolic Disease vs. NAFLD *p*-Value
Age, years	54.5 ± 7.8	56.9 ± 9.9	54.9 ± 13.0	0.05	0.73	0.19
Female, n (%)	38 (38.8)	64 (48.5)	75 (40.1)	0.14	0.80	0.15
White, n (%)	73 (74.5)	132 (100)	161 (86.1)	<0.0001	0.012	<0.0001
Type 2 diabetes, n (%)	0	76 (57.6)	96 (51.3)	<0.0001	<0.0001	<0.0001
Hypertension, n (%)	0	73 (55.3)	82 (43.9)	<0.0001	<0.0001	0.066
Hypercholesterolemia, n (%)	56 (57.1)	85 (64.4)	73 (39.0)	0.27	0.007	<0.0001
Obesity, n (%)	0	50 (37.9)	135 (72.2)	<0.0001	<0.0001	<0.0001
BMI, kg/m^2^	24.7 ± 2.6	27.9 ± 5.1	34.1 ± 6.6	<0.0001	<0.0001	<0.0001
Waist circumference, cm	-	95.6 ± 15.0	109.0 ± 14.4	-	-	<0.0001
HbA1c %	-	6.1 ± 1.1	6.4 ± 1.5	-	-	0.21
Fasting glucose, mg/dL	94.2 ± 11.5	135.5 ± 42.9	112.1 ± 40.1	<0.0001	0.0003	<0.0001
Total cholesterol, mg/dL	209.2 ± 40.1	213.2 ± 36.8	193.0 ± 45.9	0.44	0.003	<0.0001
Triglyceride, mg/dL	128.1 ± 128.1	165.1 ± 138.3	194.1 ± 123.4	0.040	<0.0001	0.07
LDL-C, mg/dL	127.2 ± 36.1	126.8 ± 35.1	112.0 ± 39.2	0.88	0.0006	<0.0001
HDL-C, mg/dL	58.4 ± 15.5	53.8 ± 16.2	45.5 ± 12.9	0.034	<0.0001	0.0008
ALT, IU/L	-	30.9 ± 16.8	74.2 ± 49.5	-	-	0.72
AST IU/L	-	26.9 ± 10.2	49.7 ± 30.6	-	-	<0.0001
ALP, IU/L	-	79.3 ± 21.8	79.5 ± 28.4	-	-	<0.0001
Total bilirubin, mg/dL	-	0.59 ± 0.30	0.58 ± 0.55	-	-	0.99
Platelet count 1k cells/μL	-	-	248.3 ± 71.8	-	-	-
Steatosis stage 2, 3, n (%)	-	-	118 (63.1)	-	-	-
Ballooning stage 1, 2, n (%)	-	-	156 (83.4)	-	-	-
Liver inflammation stage 2, 3, n (%)	-	-	69 (36.9)	-	-	-
NAS	-	-	4.5 ± 1.4	-	-	-
LSM, kPa	-	-	9.3 ± 6.5	-	-	-
Significant fibrosis (F2, F3, F4), n (%)	-	-	85 (45.5)	-	-	-
Advanced fibrosis (F3, F4), n (%)	-	-	32 (17.3)	-	-	-
Cirrhosis (F4), n (%)	-	-	14 (7.5)	-	-	-
Citrate µM	113.6 ± 28.0	115.7 ± 27.1	121.6 ± 39.6	0.54	0.050	0.009
Pyruvate, µM	42.4 ± 29.4	38.6 ± 25.7	54.0 ± 26.1	0.33	0.001	<0.0001
Total ketone bodies, µM	171.9 ± 109.7	219.7 ±38.6	229.7 ± 188.1	0.008	0.001	0.65

Values reported as mean ± standard deviation, or n (%). Abbreviations: ALP, alkaline phosphatase; ALT, alanine transaminase; AST, aspartate transaminase; BIDMC, Beth Israel Deaconess Medical Center; BMI, body mass index; HbA1c, glycated hemoglobin; HDL-C, high density lipoprotein cholesterol; LDL-C, low density lipoprotein cholesterol; LSM, liver stiffness measure (FibroScan^®^); NAFLD, nonalcoholic fatty liver disease; NAS, NAFLD activity score.

**Table 2 ijms-24-13332-t002:** Baseline clinical and laboratory characteristics of the BIDMC NAFLD cohort (n = 187) according to serum citrate >150 μM and ≤150 μM (See ref. [21] for cut-off values).

Variables	Citrate Level >150 μM (n = 38)	Citrate Level ≤150 μM (n = 149)	*p*-Value
Age, years	58.9 ± 13.8	53.9 ± 12.6	0.02
Female, n (%)	22 (57.9)	53 (35.6)	0.01
White, n (%)	34 (89.5)	127 (85.2)	0.66
High glucose *, n (%)	37 (97.4)	133 (89.3)	0.13
Type 2 diabetes, n (%)	19 (50.0)	77 (51.7)	0.85
Hypertension, n (%)	23 (60.5)	59 (39.6)	0.02
Hypercholesterolemia, n (%)	14 (36.8)	59 (39.6)	0.76
Obesity, n (%)	29 (76.3)	106 (71.1)	0.52
Metabolic syndrome, n (%)	28 (73.7)	81 (61.8)	0.18
BMI, kg/m^2^	35.5 ± 8.0	33.7 ± 6.2	0.15
Waist circumference, cm	111.3 ± 16.4	108.4 ± 13.8	0.28
HbA1c %	6.8 ± 1.7	6.2 ± 1.1	0.09
Fasting glucose, mg/dL	121.6 ± 49.2	109.1 ± 36.8	0.25
Total cholesterol, mg/dL	173.9 ± 37.7	198.1 ± 46.4	0.01
Triglycerides, mg/dL	198.6 ± 117.9	192.9 ± 125.2	0.83
LDL-C, mg/dL	95.0 ± 34.4	116.2 ± 39.4	0.01
HDL-C, mg/dL	43.5 ± 11.4	46.1 ± 13.3	0.34
ALT, IU/L	73.9 ± 40.3	74.2 ± 51.7	0.98
AST IU/L	59.2 ± 25.9	47.2 ± 31.3	0.03
ALP, IU/L	87.4 ± 33.1	77.4 ± 26.8	0.06
Total bilirubin, mg/dL	0.60 ± 0.5	0.57 ± 0.6	0.79
Platelet count, 1k cells/μL	272.0 ± 75.5	242.6 ± 69.9	0.03
Steatosis, stage 2, 3	27 (71.1)	91 (61.1)	0.25
Ballooning stage, 1, 2	37 (97.4)	119 (79.9)	0.003
Liver inflammation, stage 2, 3	19 (50.0)	50 (33.6)	0.06
NAS	5.0 ± 1.3	4.4 ± 1.4	0.01
LSM, kPa	12.8 ± 10.4	8.4 ± 4.9	0.003
Moderate fibrosis (F ≥ 2)	26 (68.4)	59 (39.6)	0.001
Advanced fibrosis (F ≥ 3)	12 (31.6)	20 (13.6)	0.01
Cirrhosis (F4, n (%)	5 (13.2)	9 (6.0)	0.16
Pyruvate, µM	65.5 ± 33.0	51.1 ± 23.3	0.002
Total ketone bodies, µM	354.2 ± 312.1	197.9 ± 123.0	<0.001

Values reported as mean ± standard deviation, or n (%). * High glucose was defined as ≥110 mg/dL as suggested by the National Cholesterol Education Program Adult Treatment Panel III (NCEP ATP III) guidelines for metabolic syndrome. Abbreviations: ALP, alkaline phosphatase; ALT, alanine transaminase; AST, aspartate transaminase; BIDMC, Beth Israel Deaconess Medical Center; BMI, body mass index; HbA1c, glycated hemoglobin; HDL-C, high density lipoprotein cholesterol; LDL-C, low density lipoprotein cholesterol; LSM, liver stiffness measure (FibroScan^®^); NAFLD, nonalcoholic fatty liver disease; NAS, NAFLD activity score.

**Table 3 ijms-24-13332-t003:** Association between liver fibrosis and clinical and laboratory variables including mitochondrial biomarkers in the NAFLD cohort (n = 187).

Variable	Univariable Association β-Coefficient and 95% CI	*p*-Value
Citrate ^†^	0.27 (0.10–0.45)	0.003
Pyruvate	0.18 (0.001–0.36)	0.049
Ketone bodies	0.08 (−0.10–0.26)	0.36
Demographics		
Age	0.02 (0.01–0.03)	0.004
Female vs. male	−0.06 (−0.43–0.30)	0.74
Comorbidities		
Obesity	0.66 (0.27–1.05)	0.001
Type 2 diabetes	0.41 (0.06–0.78)	0.02
High glucose *	0.22 (−0.44–0.88)	0.52
Hypertension	0.82 (0.48–1.16)	<0.001
High triglycerides *	−0.08 (−0.48–0.33)	0.71
Low HDL-C *	0.49 (0.09–0.88)	0.76
Increased waist circumference *	0.67 (0.24–1.10)	0.002
Metabolic syndrome *	0.94 (0.54–1.27)	<0.001
NASH severity		
NAS	0.34 (0.23–0.45)	<0.001
ALT	0.004 (0.0007–0.008)	0.02
Platelet count	−0.002 (−0.005–0.0003)	0.10

^†^ z-score for the citrate level and overall liver fibrosis as a continuous variable were used in the linear regression models. * Metabolic syndrome was defined, based on the National Cholesterol Education Program Adult Treatment Panel III (NCEP ATP III) guidelines, as three or more of the following: (1) fasting blood glucose ≥ 110 mg/dL (HbA1c ≥ 5.7% was also considered as elevated glucose equivalent), (2) hypertension and/or use of anti-hypertensive medication, (3) low HDL-C < 40 mg/dL for males and <50 mg/dL for females, (4) high TG ≥ 150 mg/dL and (5) waist circumference ≥ 102 cm for males and ≥88 cm for females. Abbreviations: ALT, alanine amino transferase; CI, confidence interval; HDL-C, high density lipoprotein cholesterol; NAFLD, nonalcoholic fatty liver disease; NAS, NAFLD activity score; NASH, nonalcoholic steatohepatitis; TG, triglycerides.

**Table 4 ijms-24-13332-t004:** Multivariable association between citrate levels and liver fibrosis in the NAFLD cohort (n = 187).

Variable	Model 1 (Age, Sex)		Model 2 (Age, Sex and Metabolic Syndrome)		Model 3 (Age, Sex, Metabolic Syndrome and NAS)	
	β-Coefficient (95% CI)	*p*-Value	β-Coefficient (95% CI)	*p*-Value	β-Coefficient (95% CI)	*p*-Value
Citrate ^†^	0.27 (0.08–0.45)	0.004	0.20 (0.02–0.38)	0.03	0.19 (0.03–0.35)	0.02
Age	0.02 (0.003–0.03)	0.02	0.01 (−0.01–0.02)	0.25	0.02 (0.003–0.03)	0.02
Female vs. male	−0.38 (−0.74–−0. 01)	0.04	−0.35 (−0.70–0.01)	0.06	−0.52 (−0.84–−0.20)	0.002
MetS			0.90 (0.53–1.27)	<0.001	0.75 (0.40–1.07)	<0.001
NAS					0.37 (0.26–0.48)	<0.001

^†^ z-score for the citrate level and overall liver fibrosis as a continuous variable were used in univariable and multivariable linear regression models. Abbreviations: CI, confidence interval; MetS, metabolic syndrome; NAFLD, nonalcoholic fatty liver disease; NAS, NAFLD activity score.

**Table 5 ijms-24-13332-t005:** Multivariable association of citrate levels with liver fibrosis in males and females in the NAFLD cohort.

Variable	Male Population (n = 112)		Female Population (n = 75)	
	β-Coefficient (95% CI)	*p*-Value	β-Coefficient (95% CI)	*p*-Value
Citrate ^†^	0.34 (0.10–0.57)	0.005	0.09 (−0.13–0.32)	0.41
Age	0.02 (0.002–0.04)	0.03	0.01 (−0.01–0.03)	0.39
Metabolic syndrome	0.57 (0.15–0.99)	0.008	0.92 (0.35–1.50)	0.002
NAS	0.35 (0.20–0.50)	<0.001	0.42 (0.25–0.59)	<0.001

^†^ z-score for the citrate level and overall liver fibrosis as a continuous variable were used in the linear regression models. Abbreviations: CI, confidence interval; NAFLD, nonalcoholic fatty liver disease; NAS, NAFLD activity score.

**Table 6 ijms-24-13332-t006:** Association of citrate levels with liver fibrosis in the NASH cohort (n = 176).

Variable	Univariable Association		Multivariable Association Adjusted for Age, Sex, NAS and Metabolic Factors	
	β-Coefficient (95% CI)	*p*-Value	β-Coefficient (95% CI)	*p*-Value
Citrate ^†^	0.23 (0.08–0.37)	0.002	0.21 (0.07–0.36)	0.005
Age	0.02 (0.01–0.04)	0.001	0.02 (0.01–0.04)	0.002
Female	−0.22 (−0.53–0.08)	0.15	−0.41 (−0.79–−0.04)	0.03
High glucose (≥110 mg/dL)	0.20 (−0.15–0.55)	0.26	0.05 (−0.30–0.39)	0.80
High triglycerides (≥150 mg/dL)	0.05 (−0.23–0.34)	0.71	0.15 (−0.12–0.42)	0.27

^†^ z-score for the citrate level and overall liver fibrosis (F0-F4) as a continuous variable were used in the linear regression models. Abbreviations: CI, confidence interval; NASH, nonalcoholic steatohepatitis; NAS, nonalcoholic fatty liver disease activity score.

## Data Availability

Data are available upon request.

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
