# Peer review of "Circulating Citrate Is Associated with Liver Fibrosis in Nonalcoholic Fatty Liver Disease and Nonalcoholic Steatohepatitis"

_ijms, 2023, doi:10.3390/ijms241713332_

Round 1

Reviewer 1 Report

This was a very well-written and comprehensive paper investigating serum citrate levels and NAFLD. The authors concluded that there was an association between serum citrate levels and fibrosis in males.  In light of these results, it may be helpful to spilt Figure 2 A into two graphs: one for females and one for males. I suggest this because, in two parts of the paper (255-258 and 273-277), there is a discussion of females having higher citrate levels but not males.  Can you expand on this a bit?

Please define high glucose in tables 2 and 3.  

There is one minor typo on line 128: F is missing from the 4, describing the advanced fibrosis.

Author Response

This was a very well-written and comprehensive paper investigating serum citrate levels and NAFLD.

The authors would like to thank the reviewer for their kind words and for the thoughtful review of our manuscript.

The authors concluded that there was an association between serum citrate levels and fibrosis in males.  In light of these results, it may be helpful to spilt Figure 2 A into two graphs: one for females and one for males. I suggest this because, in two parts of the paper (255-258 and 273-277), there is a discussion of females having higher citrate levels but not males.  Can you expand on this a bit?

Thank you for this suggestion.  Figure 2 was amended by adding separate graphs for males and females. A full paragraph expanding on the observation that citrate levels were higher in females than males can be found in the discussion section (see lines 255-280).

Please define high glucose in tables 2 and 3.  

High glucose was defined as ≥110 mg/dL as suggested by the NCEP ATP III guidelines (see methods section—lines 361-367). This value as well as the rest of the definitions for the risk factors for metabolic syndrome were added to the footnotes of tables 2 and 3 for clarity.

There is one minor typo on line 128: F is missing from the 4, describing the advanced fibrosis.

Great catch!  This mistake has been fixed.

Reviewer 2 Report

I read the manuscript entitled "Circulating Citrate is Associated with Liver Fibrosis in Nonalcoholic Fatty Liver Disease and Nonalcoholic Steatohepatitis" with great interest. The research design is appropriate, and the methods are adequately described. There are just some points to resolve.

In line 37, you mention MASLD according to the most recent nomenclature. However, you should briefly describe the metabolic impairments better known as MAFLD (see here: doi.org/10.1016/j.jhep.2020.03.039).

Lines 50–52: "This premise is further supported by observations that metabolites of mitochondrial activity, such as circulating citrate and intrahepatocyte β-hydroxybutyrate, are elevated in NASH [8,10]". This is crucial, but you should provide more strong evidence, such as human data from clinical trials (one study includes 6 subjects, another one talks about Multi-Omics Data from mice, and the last one is a short communication).

Lines 76–77: "High-fat diets and MetS both confer aconitase inhibition, which conceivably contributes to the elevation of circulating citrate [13,14]" This is partially true because alteration of general omeostatic mechanisms leads to impaired aconitase activity. Nevertheless, in the case of high-fat diets, there are some exceptions (e.g., the ketogenic diet; see here: doi:10.1002/adbi.202300095).

Lines 84–87: Add more references related to the role of mithocondria and dysfunction linked to liver damage since many strategies to prevent or restore liver function focus on the improvement of mitochondrial activities (see doi: 10.1053/j.gastro.2018.06.083, doi.org/10.3390/antiox12051065).

Line 146: Explain what you mean by "Apparently healthy participants."

Line 338-342: add some refs

In the results, in table 2, it’s very interesting to note the very high value of ketone bodies (KBs). Convert KBs to mmol/L and remember that over 3 mmol/L is very high and means you may have Diabetic ketoacidosis (https://www.nhs.uk/conditions/diabetic-ketoacidosis/). So, beyond the entire group of MetS, focus on the dataset of diabetic subjects.

Line 407: Citrate is, of course, a fundamental substrate for cellular metabolism in different energy processes; differently from HCO3−, citrate can penetrate the sarcolemma (add ref doi:10.1002/tsm2.174), so you should underline the cause of the problem (altered omeostasis, such as in diabetics, for example) and not the consequence (elevated citrate levels).

Overall, the article is interesting, well written, and organised.

Author Response

I read the manuscript entitled "Circulating Citrate is Associated with Liver Fibrosis in Nonalcoholic Fatty Liver Disease and Nonalcoholic Steatohepatitis" with great interest. The research design is appropriate, and the methods are adequately described. There are just some points to resolve.

The authors would like to thank you for taking the time to thoroughly review our manuscript and for your thoughtful suggestions that have greatly improved our manuscript.

In line 37, you mention MASLD according to the most recent nomenclature. However, you should briefly describe the metabolic impairments better known as MAFLD (see here: doi.org/10.1016/j.jhep.2020.03.039).

Thank you for pointing out this manuscript. It was our mistake to overlook this preceding publication that highlights the international definition of MAFLD (see lines 38-42).  We have now added a sentence to this effect and this reference to the introduction (see new reference 3).

Lines 50–52: "This premise is further supported by observations that metabolites of mitochondrial activity, such as circulating citrate and intrahepatocyte β-hydroxybutyrate, are elevated in NASH [8,10]". This is crucial, but you should provide more strong evidence, such as human data from clinical trials (one study includes 6 subjects, another one talks about Multi-Omics Data from mice, and the last one is a short communication).

We agree that the previous data was not strong which is why we decided to conduct this study.  We believe that the results of our study provide strong evidence for these earlier observations.

Lines 76–77: "High-fat diets and MetS both confer aconitase inhibition, which conceivably contributes to the elevation of circulating citrate [13,14]" This is partially true because alteration of general omeostatic mechanisms leads to impaired aconitase activity. Nevertheless, in the case of high-fat diets, there are some exceptions (e.g., the ketogenic diet; see here: doi:10.1002/adbi.202300095).

Thank you for pointing out this reference.  We have added the following proviso to the above mentioned sentence.  The sentence now reads: “With the exception of some ketogenic diets, high-fat diets and MetS both confer aconitase inhibition, which conceivably contributes to the elevation of circulating citrate [14-16]” (see lines 83-85). The reference was also added (see new reference 16).

Lines 84–87: Add more references related to the role of mithocondria and dysfunction linked to liver damage since many strategies to prevent or restore liver function focus on the improvement of mitochondrial activities (see doi: 10.1053/j.gastro.2018.06.083, doi.org/10.3390/antiox12051065).

These two references were added as suggested (lines 91-92) (see new references 20 and 21).

Line 146: Explain what you mean by "Apparently healthy participants."

It is common to say “apparently healthy participants” when you have not completely ruled out any possible underlying disease in subjects that are being recruited for a control/reference cohort. So as to be less confusing, we have replaced the term “healthy” and now call it the “control cohort.”

Line 338-342: add some refs

References were added as suggested.

In the results, in table 2, it’s very interesting to note the very high value of ketone bodies (KBs). Convert KBs to mmol/L and remember that over 3 mmol/L is very high and means you may have Diabetic ketoacidosis (https://www.nhs.uk/conditions/diabetic-ketoacidosis/). So, beyond the entire group of MetS, focus on the dataset of diabetic subjects.

If converted from µmol/L (µM) to mmol/L, the concentration of total ketone bodies in the subjects with NAFLD was 0.23 mmol/L which is much lower than the levels of about 3 mmol/L that are typically observed in diabetic ketoacidosis.  We thought this was worth mentioning in the text so we added a sentence in the results section that reads: “Despite being higher than normal, the mean ketone body concentration in the NAFLD cohort (0.23 mmol/L) was much lower than one would experience with diabetic ketoacidosis (>3 mmol/L).” (lines 120-123)

Line 407: Citrate is, of course, a fundamental substrate for cellular metabolism in different energy processes; differently from HCO3−, citrate can penetrate the sarcolemma (add ref doi:10.1002/tsm2.174), so you should underline the cause of the problem (altered omeostasis, such as in diabetics, for example) and not the consequence (elevated citrate levels).

Thank you for pointing out this relevant paper.  To address this issue, we added the following sentences to the discussion section: “It is hard, however, to tell for sure if the increase in circulating citrate, which is presumably caused by the increase in substrates for the TCA cycle, is simply a consequence of obesity, T2D and/or NAFLD or if it contributes to the loss in mitochondrial homeostasis. Additionally, because citrate can penetrate the sarcolemma and is a fundamental substrate for cellular metabolism in several different energy processes, there is evidence that taking sodium citrate as a dietary supplement could be beneficial [39]. However, it is unknown whether higher citrate levels represent a beneficial adaptation or if citrate may exacerbate the underlying disease state. More work needs to be done to understand the cause and effect of these observations.” (lines 302-311)

Overall, the article is interesting, well written, and organised.

The authors very much appreciate your kind words as we tried to tackle a fairly difficult subject.

Round 2

Reviewer 2 Report

Kudos to all authors